# Newly Synthesized Oxygenated Xanthones as Potential P-Glycoprotein Activators: *In Vitro*, *Ex Vivo*, and *In Silico* Studies

**DOI:** 10.3390/molecules24040707

**Published:** 2019-02-15

**Authors:** Eva Martins, Vera Silva, Agostinho Lemos, Andreia Palmeira, Ploenthip Puthongking, Emília Sousa, Carolina Rocha-Pereira, Carolina I. Ghanem, Helena Carmo, Fernando Remião, Renata Silva

**Affiliations:** 1UCIBIO-REQUIMTE, Laboratório de Toxicologia, Departamento de Ciências Biológicas, Faculdade de Farmácia, Universidade do Porto, Rua de Jorge Viterbo Ferreira 228, 4050-313 Porto, Portugal; evagilmartins18@gmail.com (E.M.); veralssilva17@gmail.com (V.S.); mcamorim@ff.up.pt (C.R.-P.); helenacarmo@ff.up.pt (H.C.); 2Laboratório de Química Orgânica e Farmacêutica, Departamento de Ciências Químicas, Faculdade de Farmácia, Universidade do Porto, Rua Jorge Viterbo Ferreira 228, 4050-313 Porto, Portugal; up201002662@ff.up.pt (A.L.); apalmeira@ff.up.pt (A.P.); 3Faculty of Pharmaceutical Sciences, Khon Kaen University, Khon Kaen 40002, Thailand; pploenthip@gmail.com; 4Centro Interdisciplinar de Investigação Marinha e Ambiental (CIIMAR/CIMAR), Universidade do Porto, Rua dos Bragas 289, 4050-123 Porto, Portugal; 5Universidad de Buenos Aires, CONICET, Facultad de Farmacia y Bioquímica, Instituto de Investigaciones Farmacológicas (ININFA), Buenos Aires C1053, Argentina; cghanem@ffyb.uba.ar

**Keywords:** P-glycoprotein, induction, activation, oxygenated xanthones, intestinal barrier, intoxication scenarios

## Abstract

P-glycoprotein (P-gp) plays a crucial role in the protection of susceptible organs, by significantly decreasing the absorption/distribution of harmful xenobiotics and, consequently, their toxicity. Therefore, P-gp has been proposed as a potential antidotal pathway, when activated and/or induced. Knowing that xanthones are known to interact with P-gp, the main goal was to study P-gp induction or/and activation by six new oxygenated xanthones (OX 1-6). Furthermore, the potential protection of Caco-2 cells against paraquat cytotoxicity was also assessed. The most promising compound was further tested for its ability to increase P-gp activity *ex vivo*, using everted intestinal sacs from adult Wistar-Han rats. The oxygenated xanthones interacted with P-gp *in vitro*, increasing P-gp expression and/or activity 24 h after exposure. Additionally, after a short-incubation period, several xanthones were identified as P-gp activators, as they immediately increased P-gp activity. Moreover, some xanthones decreased PQ cytotoxicity towards Caco-2 cells, an effect prevented under P-gp inhibition. *Ex vivo*, a significant increase in P-gp activity was observed in the presence of OX6, which was selectively blocked by a model P-gp inhibitor, zosuquidar, confirming the *in vitro* results. Docking simulations between a validated P-gp model and the tested xanthones predicted these interactions, and these compounds also fitted onto previously described P-gp induction and activation pharmacophores. In conclusion, the *in vitro*, *ex vivo*, and *in silico* results suggest the potential of some of the oxygenated xanthones in the modulation of P-gp, disclosing new perspectives in the therapeutics of intoxications by P-gp substrates.

## 1. Introduction

P-glycoprotein (P-gp) is the best characterized efflux protein pump of the adenosine triphosphate (ATP) binding cassette (ABC) superfamily of transporters, belonging to the ABC subfamily B (ABCB) [1,2,3]. P-gp was first discovered and isolated from Chinese hamster ovary cells by Juliano and Ling in 1976 [4]. It is a well-studied protein due to its importance in the protection of sensitive tissues against toxic xenobiotics, although this mechanism it is not yet fully understood, and to its important role in the multi-drug-resistance (MDR) phenomenon in cancer chemotherapy [1,4,5].

In humans, P-gp is encoded by two multidrug resistance (MDR) genes, *MDR1/ABCB1*, coding to the drug transporter associated with the MDR phenotype, and *MDR3*/*ABCB4* (or *MDR2*), coding to a protein that functions as a phosphatidylcholine translocase, exporting this phospholipid into the bile [6].

It is believed that this 170 kDa protein is a result of a gene duplication event, where two transmembrane domains (TMDs), which usually contain six transmembrane α-helices (TMHs) and one nucleotide binding domain (NBD) each, are fused together [1,7]. NBD regions, located in the cytoplasmic membrane side, bind and hydrolase ATP producing energy to execute the membrane passage, with the TMHs responsible for the pathway of this passage [3,7].

P-gp has a low expression in most human tissues [8], although it is found in much higher concentrations in the apical surface of epithelial cells of the intestine, liver bile ductules, kidney proximal tubules, pancreatic ductules, adrenal gland, placenta, pregnant endometrium, testis, capillary endothelial cells of the cochlea and vestibule, and in the blood-brain barrier [7,8,9,10,11,12,13,14,15,16]. P-gp was also found on the surface of hematopoietic cells, but its function remains unknown [17]. In general, P-gp tissue localizations suggest that this protein is associated with the protection of susceptible organs against toxic xenobiotics, as has already been proven in many different studies [6,7,18].

P-gp can export a large number of structurally different compounds, including chemotherapeutic drugs, steroids, and natural products, among others. These compounds are usually amphipathic, relatively hydrophobic, and nonpolar [7,16,19].

In previous studies, P-gp’s capacity to work in the protection of tissues or/and cells, by limiting the intracellular accumulation of its substrates, like paraquat (PQ), and consequently reducing their toxicity, was hypothesized and many strategies to increase and/or activate P-gp expression and activity were developed. P-gp activators have the benefit of increasing P-gp activity without interfering with P-gp protein expression, conferring a higher speed of action when compared with the use of P-gp inducers [2,5,7,20].

In order to discover a potential new antidotal pathway to be used in intoxications by toxic P-gp substrates through the increase in P-gp-mediated efflux, PQ was used as a model of a toxic P-gp substrate [20,21], and newly synthesized xanthonic derivatives were used as potential P-gp inducers and/or activators. These molecules have a vast list of known benefits, such as cardiovascular, antimicrobial, antitumor, and anticonvulsant properties, among others [22,23], evoking special interest amongst medicinal chemistry investigators that considered these compounds as “privileged structures” [24], mostly due to their interaction ability with a diverse range of biomolecules. Nevertheless, the xanthone—drug transporter interaction mechanism remains sparse [5].

Very recently, a simple oxygenated carbaldehyde xanthone was discovered as the first activator of TAp73 [25], prompting the study of their analogues [25]. In this work, we thus aimed to evaluate the effect of a series of six newly synthesized oxygenated xanthones (OXs) (Figure 1) on P-gp expression and activity, and their potential to protect Caco-2 cells against PQ-mediated cytotoxicity, which is an herbicide with a very high mortality rate in cases of intoxication. A recent study already demonstrated that a group of dihydroxylated xanthonic derivatives efficiently induced and activated P-gp, conferring protection against this toxic substrate [5]. In the present study, the type of substituents was diversified to alkyl, halogenated, aldehyde, and alcohol to understand their influence on P-gp modulation, and Caco-2 cells were used as an *in vitro* model for this purpose, since these cells were described to express P-gp at similar levels to that observed in a normal human jejunum and were already validated for the screening of P-gp inducers and activators [5,21,26,27].

Additionally, using everted intestinal sacs of Wistar-Han rats, the effect of the most promising compound on P-gp activity was evaluated *ex vivo*. Also, docking studies were performed to better understand the newly synthesized xanthones’ mode of action and interaction with P-gp.

## 2. Results

### 2.1. Synthesis of the Oxygenated Xanthones (OXs)

The commercially available 1,2,3-trimethoxy-5-methylbenzene was implemented as the starting precursor for the preparation of 3,4-dimethoxy-1-methyl-9*H*-xanthen-9-one (OX1) through Friedel-Crafts acylation with 2-methoxybenzoyl chloride and subsequent alkaline cyclization under microwave irradiation. This two-step methodology has been described in the literature [23,28,29]. Benzylic dibromination of OX1 using dibenzoyl peroxide (DBP) and *N*-bromosuccinimide (NBS) afforded the 1-(dibromomethyl)-3,4-dimethoxy-9*H*-xanthen-9-one (OX2).

The synthesis of 3,6-dihydroxy-9-oxo-9*H*-xanthene-4,5-dicarbaldehyde (OX3) and 4-hydroxy-3-methoxy-9-oxo-9*H*-xanthene-1-carbaldehyde (OX4) was achieved via Duff formylation of 3,6-dihydroxy-9*H*-xanthen-9-one and 4-hydroxy-3-methoxy-9*H*-xanthen-9-one, respectively, using hexamethylenetetramine (HMTA) in acetic acid [30,31]. Methylation of the hydroxy group of OX4 with dimethylsulfate and potassium carbonate furnished the 3,4-dimethoxy-9-oxo-9*H*-xanthene-1-carbaldehyde (OX5) [25].

1-(Hydroxymethyl)-3,4-dimethoxy-9*H*-xanthen-9-one (OX6) was synthesized via reduction of the xanthone OX5 with Biotage^®^ MP-Borohydride (MP-BH_4_) using a previously reported methodology [25].

The purity of each synthesized xanthone was determined by high performance liquid chromatography with diode-array detection (HPLC–DAD) analysis. All tested compounds presented a purity of at least 95%.

### 2.2. Oxygenated Xanthones Cytotoxicity Assays

Xanthones’ (OX 1–6) cytotoxicity was evaluated by the neutral red (NR) uptake assay, 24 h after exposure, to select a noncytotoxic working concentration. As shown in Figure 2, for all the compounds under study, no significant cytotoxicity was detected for any of the tested concentrations (0–50.0 μM) and up to 24 h of exposure, when compared to control cells (C, 100%). Accordingly, the concentration of 20.0 μM was selected for the subsequent experiments, since this noncytotoxic concentration was already reported, for similar xanthonic derivatives, as a concentration able to cause a significant increase in both P-gp expression and activity [2,5,27].

### 2.3. Evaluation of P-Glycoprotein Expression

The effect of the tested xanthones on P-gp expression was evaluated by flow cytometry, using a P-gp monoclonal antibody (UIC2) conjugated with phycoerythrin (PE). As shown in Figure 3, the tested xanthone, OX4, significantly increased P-gp expression to 115%, as compared with control cells (C, 100%). On the other hand, the tested xanthone, OX1, significantly reduced P-gp expression to 79%, when compared to control cells (C, 100%). For OX2, OX3, OX5, and OX6, no significant effect on P-gp expression was observed, when compared to control cells (C, 100%). Representative histograms of the flow cytometry analysis of Caco-2 cells’ autofluorescence and UIC2-PE binding after exposure to OX4 for 24 h are illustrated in Appendix A, respectively (Appendix A).

### 2.4. Evaluation of P-Glycoprotein Transport Activity

P-gp activity was evaluated using two different protocols: The first approach, by evaluating the accumulation of Rhodamine 123 (Rho 123) in the presence of the OXs (20.0 μM) during the 60-min accumulation phase of the fluorescent P-gp substrate (Figure 4); and the second approach, by evaluating the accumulation of Rho 123 after pre-exposure of Caco-2 cells to the OXs (20.0 μM) for 24 h (Figure 5). The first assay aims to evaluate the potential immediate effect of the tested xanthonic derivatives on P-gp activity, as a direct result of the pump activation. On the other hand, the second assay allows an evaluation of whether the potential increases in P-gp expression result in an increase in its activity. In both cases, the incubations were performed in the presence and absence of the P-gp inhibitor, Elacridar (Ela, 10.0 μM).

As observed in Figure 4, all the tested oxygenated xanthones except xanthone OX3, significantly and immediately (given the short incubation period) increased P-gp activity when compared to control cells (C, 100%). P-gp activity increased to 130%, 130%, 121%, 130%, and 132% when the accumulation of Rho 123 was assessed in the presence of the xanthonic derivatives, OX1, OX2, OX4, OX5, and OX6, respectively. Representative histograms of the flow cytometry analysis of Caco-2 cells’ autofluorescence and Rho 123 fluorescence after exposure to OX6 during the accumulation of the fluorescent substrate are illustrated in Appendix A, respectively (Appendix A).

Considering the second experimental protocol (Figure 5), it is possible to verify that the xanthonic derivatives, OX1, OX2, OX5, and OX6, significantly increased P-gp activity in comparison to control cells (C, 100%). P-gp activity increased to 132%, 123%, 128%, and 126% in Caco-2 cells pre-exposed for 24 h to the oxygenated xanthones, OX1, OX2, OX5, and OX6, respectively. For xanthones OX3 and OX4, no significant increase in P-gp activity was observed, when compared to control cells (C, 100%). Representative histograms of the flow cytometry analysis of Caco-2 cells’ autofluorescence and Rho 123 fluorescence after exposure to OX5 for 24 h are illustrated in Appendix A, respectively (Appendix A).

### 2.5. Oxygenated Xanthones’ Protective Effects Against Paraquat-Induced Cytotoxicity

Paraquat (PQ) is a well-known herbicide and a very toxic P-gp substrate. To verify a possible xanthone-mediated protective effect on PQ-induced toxicity, the herbicide cytotoxicity (0–5000.0 μM) was evaluated with or without simultaneous exposure to the herbicide and the tested oxygenated xanthones (OX 1–6, 20.0 μM). PQ cytotoxicity was evaluated by the NR uptake assay after 24 h of incubation. P-gp positive modulation may result in a significant increase in P-gp-mediated efflux, and, consequently, in a reduction in PQ cytotoxicity. Figure 6 shows the concentration—response curves obtained with only PQ incubation and with simultaneous incubation with the herbicide and the tested oxygenated xanthones (PQ + OX 1–6). As exposed in Figure 6, for the xanthones, OX2, OX3, OX4, and OX5, a significant reduction in cell death was observed for the 250.0–2500.0 μM PQ concentration range, resulting in a significant rightward shift of all the PQ + OXs curves, when compared with the PQ curve alone. Furthermore, after simultaneous exposure to the xanthones, OX2, OX3, OX4, and OX5, no significant differences were perceived neither in the maximal cell death (top) nor in the baseline (bottom) of the fitted curves, when compared to the PQ curve alone (Table 1). For that reason, the EC_50_ values, which represent the half-maximum-effect concentrations from the fitted curves, were used for comparisons. As shown in Table 1, the simultaneous exposure to PQ and the xanthones, OX2, OX3, OX4, and OX5, resulted in a significant increase in the EC_50_ value of the fitted curves, when compared to the EC_50_ of the PQ curve alone (1153, 1291, 1188, and 1217 μM for the PQ + OX2, PQ + OX3, PQ + OX4, and PQ + OX5 curves, respectively, when compared with the EC_50_ of the PQ curve (982.4 μM)). Nonetheless, after simultaneous exposure to PQ and the xanthones, OX1 and OX6, no significant differences were observed between the PQ and the PQ + OX1 and PQ + OX6 curves, demonstrating a lack of protection against PQ-mediated cytotoxicity. According to the obtained results, xanthone OX3 was demonstrated to be the most active compound, causing a significant decrease in PQ-induced toxicity towards Caco-2 cells.

To further investigate if the observed protective effects against PQ-induced cytotoxicity were mediated by P-gp, the previous experiment was repeated in the presence of a specific P-gp inhibitor, Ela, at a non-cytotoxic concentration of 10.0 μM (Appendix A). According to the obtained results, PQ cytotoxicity significantly increased in the presence of Ela, resulting in a significant leftward shift of the PQ + Ela curve, when compared with the PQ curve alone (Appendix A). Consequently, a significant decrease in the EC_50_ value of the PQ + Ela curve was detected, thus P-gp inhibition had a significant impact on PQ cytotoxicity (450.3 μM for the PQ curve when compared with the EC_50_ of the PQ curve (982.4 μM)) (Appendix A).

Additionally, in Figure 7, under P-gp inhibition, a complete abolishment of the xanthones’, OX1, OX4, and OX5, protective effect against PQ-induced toxicity was observed. For the tested xanthonic derivatives, OX1, OX4, and OX5, a leftward shift of the PQ + Ela + OXs curves was noticed, when compared with the PQ + Ela curve alone, thereby resulting in significant alterations in the overall comparison of the fitted curves (Table 2). Furthermore, for the xanthones, OX4 and OX5, a significant increase in cell death was detected with P-gp inhibition (PQ + Ela + OXs) for the 50.0–250.0 μM PQ concentration range when compared with P-gp inhibition alone (PQ + Ela) (Figure 7). Though, as shown in Table 2, no significant differences were obtained in the corresponding EC_50_ values. For the remaining tested oxygenated xanthones (OX1, OX2, OX3, and OX6), no significant differences were detected neither in the overall comparison of the fitted curves (Figure 7) nor in the comparison of individual parameters (EC_50_, TOP, BOTTOM, and Hill slope, Table 2).

### 2.6. Ex Vivo Studies—Evaluation of P-Glycoprotein Transport Activity in Rat Everted Intestinal Sacs

P-gp is highly expressed at the apical membrane of enterocytes, promoting the efflux of compounds from the cells back into the intestinal lumen [32]. The expression of P-gp along the intestine is not uniform, as it has high levels in the ileum and colon, in contrast to the jejunum and duodenum, where its expression is lower [32]. The *ex vivo* studies aimed to evaluate the modulatory effect of one of the tested compounds over P-gp expressed at the apical membrane of enterocytes from the distal portion of the Wistar-Han rat intestine. OX6 was selected based on the *in vitro* evaluation of the effect of the oxygenated xanthones on P-gp transport activity. To achieve this purpose, the distal portion of the ileum (portion where P-gp expression is the highest) was removed and everted to prepare the intestinal sacs. After inversion, the P-gp expressed on the apical membrane of the enterocytes was turned towards the sac exterior, in close contact with the Krebs-Henseleit (KH) medium, promoting the transport of the fluorescent substrate, Rho 123, placed inside the sac (serosal to mucosal transport). Thus, the serosal to mucosal transport of Rho 123 was evaluated, in the presence or absence of the oxygenated xanthone, OX6, by spectrofluorometry, in all medium samples. Figure 8 shows that xanthone OX6 was able to significantly increase the cumulative efflux of Rho 123, when compared to the control everted intestinal sacs. This tendency to increase the serosal to mucosal transport of the fluorescent substrate was present since minute 5, however, the significant differences occurred from minute 25 onwards, when the controls and OX6-treated sacs were compared. The Rho 123 concentration at the mucosal side of the intestinal membrane significantly increased from 12.6 ± 3.4, 16.8 ± 4.9, 21.8 ± 6.1, 30.4 ± 5.9, and 32.2 ± 7.6 in the control everted intestinal sacs to 24.1 ± 4.4, 31.6 ± 4.7, 38.8 ± 5.4, 46.7 ± 6.9, and 54.1 ± 6.8 pmol/mg tissue in the presence of the xanthone, OX6, at sampling times of 25 to 45 min, respectively. Furthermore, this increase was blocked by zosuquidar (Zos), a specific P-gp inhibitor, which proves P-gp involvement in the observed increase in the efflux of Rho 123.

### 2.7. In Silico Studies

As previously mentioned, P-gp is an ATP binding cassette transporter formed by four domains: Two TMDs (TMD1 and TMD2, each one composed of six spanning alfa-helixes), which contain the drug binding sites and define the translocation pathway across the membrane; and two cytoplasmic NBDs (NBD1 and NBD2), which bind and hydrolyse ATP, and this energy allows drug transport [33,34] (Figure 9).

The 3.4 Å resolution structure of mouse P-gp (pdb code 4Q9H) was selected for the molecular modelling study [35]. The backbone root-mean-squared error displacement (RMSD) between the mouse P-gp (pdb code 4Q9H) and the obtained P-gp model was 0.194 Å after fitting the backbone atoms. The quality of the entire P-gp model was assessed on a single point energy calculation (the final total energy of the model was −29,242 kcal mol^−1^), by comparison with the target, 4Q9H, and by analysis of the Ramachandran plot (98.7%, i.e., 1179/1194 of all residues were in the allowed regions).

Docking studies were performed on both the TMD and NBD, using known P-gp inhibitors and activators as positive controls [36,37,38] (Table 3 and Figure 9).

Amongst the tested compounds, OX2 bound with P-gp TMD with the highest affinity (highly negative docking scores, Table 3), presenting values of free energy more negative then known P-gp inhibitors, such as verapamil, and equal to the second generation P-gp inhibitor, biricodar. Besides, OX2’s free energy of binding was similar, but higher than the known P-gp activators (Table 3). A visual inspection of the known inhibitors and the tested xanthones on the TMD and NBD, and known activators and tested xanthones on the TMD was performed.

Figure 9B reveals that OX2, which presented the most negative docking score on the P-gp TMD, binds at the same drug-binding pocket as known inhibitors, such as verapamil (Figure 9B), and known activators, such as coelenteramide (Figure 9C). Both verapamil and OX2 establish π-stacking interactions with Phe-938, and polar interactions with Asp-188 and Lys-934, respectively (Figure 9B). Coelenteramide establishes polar interactions with Glu-875 and Thr-941, as well as π-stacking interactions with Phe-938 (Figure 9C). OX2, which presented the most negative docking score on the P-gp NBD amongst the tested small molecules, binds in the same site as known P-gp ATPase inhibitors, such as baicalein (Figure 9D). The residues involved in polar interaction with OX2 and baicalein are Asp-164 and Arg-905, respectively; both molecules establish π-stacking interactions with Tyr-401 (Figure 9D). *In silico* studies evaluating the interactions between xanthones and P-gp have already been performed by our group [5]. However, by that time, the most suitable target to build a human P-gp model was Sav1866, an ATP-binding cassette protein from *Staphylococcus aureus*, with a 30% sequence similarity and a TMD outward-facing conformation. Despite the structural differences of that model and the model described herein, xanthones had already been predicted to establish hydrogen interactions on a binding site engulfed by TMD 10.

In order to evaluate the potential induction and activation mechanism of the tested oxygenated xanthones, they were matched to previously described P-gp induction [26] (Figure 10A,B, Table 4, pharmacophores I, III, and IV) and P-gp activation [2] (Figure 10C, Table 4, pharmacophore V) pharmacophores [26], and their fitting scores were evaluated.

The oxygenated xanthone, OX3, was the molecule with the best fitting to pharmacophore I, although a divergence to the ideal feature positions led to a relatively low score (Figure 10A, Table 4). All the molecules had a fit score inferior to two on pharmacophore III, which was considered as non-matching results. The best matches were obtained for the compounds, OX6 and OX4, on pharmacophores IV and V, respectively. All the compounds missed one of the three features of pharmacophore II, and therefore this is considered a non-matching result (not shown). Accordingly, OX6, OX3, and other xanthones demonstrated their potential to be P-gp inducers. Compound OX4 is predicted as being the most active P-gp activator as it presents pharmacophoric fit values of 2.99 in 3.00 (Figure 10C, Table 4, Pharmacophore V).

## 3. Discussion

The present data clearly demonstrated the potential of the tested oxygenated xanthones, at a non-cytotoxic concentration of 20.0 µM (OX 1–6), to significantly increase P-gp expression (xanthone OX4) and activity (xanthones OX1, OX2, OX5, and OX6), 24 h after exposure. Furthermore, after a short incubation period of 60 min, the xanthones, OX1, OX2, OX4, OX5, and OX6, efficiently and immediately increased P-gp activity, without changing the protein expression, thus suggesting that these compounds acted as P-gp activators. Additionally, some of the tested xanthones (OX2, OX3, OX4, and OX5) significantly protected Caco-2 cells against PQ-induced cytotoxicity. *Ex vivo* studies also confirmed the ability of xanthone OX6 to significantly increase the efflux of Rho 123, in rat intestinal everted sacs, confirming the *in vitro* results. Docking studies on a human P-gp model and analysis of the fitting scores on P-gp activation and induction pharmacophores give a glimpse of the potential mechanism of action of the tested xanthones.

Caco-2 cells, the *in vitro* model used in these experiments was previously validated as a suitable model for the screening of several P-gp modulators, including xanthonic and thioxanthonic derivatives [2,5,21,26,27]. Silva et al. conducted two studies in Caco-2 cells, where five newly dehydroxylated xanthones and five thioxanthonic derivatives were screened as potential modulators of P-gp expression and activity. They found a significant increase in P-gp expression and activity in cells exposed to all tested (thio)xanthonic derivatives for 24 h, thus behaving as P-gp inducers. Additionally, these compounds also rapidly increased P-gp activity without increasing its expression, showing, therefore, a P-gp activation behaviour. In addition, these studies also evaluated the potential protective effects of (thio)xanthones against the cytotoxicity caused by the harmful P-gp substrate, PQ. All dehydroxylated xanthones and four out of five thioxanthonic derivatives were found to significantly reduce PQ cytotoxicity, highlighting the relevance of P-gp modulation in cell protection against toxic substrates [2,5]. Using the same *in vitro* model, Lopes et al. evaluated the capacity of four enantiomeric pairs of newly synthesized chiral aminated thioxanthones to modulate P-gp expression and/or activity, revealing that all the tested compounds immediately (simultaneous incubation of the tested xanthones and Rho 123) increased, in a significant manner, the P-gp activity, when compared to control cells (between 114% and 132%, when compared to control cells (100%)). Most of the compounds also increased P-gp activity after a pre-incubation with the tested xanthones for 24 h (between 147% and 176%, when compared to control cells (100%)). However, this increase in the efflux pump activity was not a reflection of an increased P-gp expression, since only one thioxanthone significantly increased P-gp expression by 36%, when compared to control cells, and after 24 h of incubation [27]. Therefore, these results highlight that P-gp activity can be increased without an increase in P-gp expression, reinforcing the importance of simultaneous evaluation of P-gp expression and activity.

In the present study, only xanthone OX4 significantly increased P-gp expression by 15%, although no significant increase in P-gp activity was observed in Caco-2 cells pre-exposed for 24 h to this compound. Thus, even though the *de novo* synthesized P-gp is already incorporated into the plasma membrane (since the UIC2 antibody recognizes an external epitope of the protein), it may not yet be fully functional. On the other hand, the increase in P-gp activity observed after 24 h of pre-exposure to the xanthones, OX1, OX2, OX5, and OX6, does not result from an increase in the pump expression, since no significant increase in cell-surface P-gp expression was observed 24 h after the exposure to these compounds. However, these xanthones (OX1, OX2, OX5, and OX6) demonstrated their ability to immediately increase P-gp activity, behaving as P-gp activators. Therefore, the increased activity observed in Caco-2 cells pre-exposed to these compounds for 24 h may be the result of a direct activation of the pump caused by the compound that may have remained intracellularly after the exposure period.

The results obtained concerning the evaluation of the OXs’ effects on P-gp expression and activity further reinforce that this approach of evaluation of both parameters has an important impact on the evaluation of the modulatory potential of new compounds. Furthermore, the data is in accordance with other reports that also suggest that alterations in P-gp expression do not linearly translate into similar changes in P-gp transport activity (which could be due to an insufficient time to guarantee complete protein function or incomplete membrane targeting) and increases in P-gp activity may occur without changes in protein expression [21,26,27,39]. In fact, Silva et al. observed in Caco-2 cells that the strong increases in P-gp expression levels after exposure to doxorubicin (Dox) (reaching 646% after 24 h of incubation with 100 μM Dox when compared to control cells (100%)) were not translated into proportional increases in P-gp transport activity (150% after a 24 h incubation period with 100 μM Dox, when compared to control cells (100%)) [21].

In accordance, Wongwanakul et al. found that treatment of Caco-2 cells with 3 μM Dox for 1 and 7 days, significantly increased P-g expression up to 2.15-fold and 3.76-fold over control cells, respectively. However, P-gp activity did not accompanied these increases in protein expression, revealing no significant alterations at any of the tested Dox concentrations (1, 3, and 10 μM) and up to 7 days of incubation [40]. In the same study, short-term exposure of Caco-2 cells to rhinacanthin-C (100 μM) for 1 day also resulted in a significant increase in P-gp expression up to 1.74 fold of the control cells, without any significant change in its function. Also, Vilas-Boas et al. found that the age-dependent increase in P-gp expression observed in human lymphocytes isolated from whole blood samples was not accompanied by a correspondent increase in protein transport activity, reinforcing the need to assess both P-gp expression and activity [39]. Additionally, Silva et al, using Caco-2 cells as an *in vitro* model, further demonstrated a concentration-dependent increase in P-gp expression induced by colchicine, a known P-gp substrate and inducer (129%, 135%, 145%, 150%, 154%, and 183% after 24 h of exposure to 0.5, 1, 5, 10, 50, and 100 μM colchicine, correspondingly, when compared to control cells (100%)) with no significant changes in the protein transport activity. These results also suggest that P-gp incorporation into the cell membrane does not guarantee protein functionality, with the lack of an effect of colchicine on P-gp activity explained by its potential action as a P-gp competitive inhibitor, as supported by the performed *in silico* studies [26]. Furthermore, knowing that the success of adopting P-gp induction as a potential therapeutic approach, both in intoxications by toxic xenobiotics that are P-gp substrates and in the reduction of the accumulation of toxic endogenous compounds, is remarkably dependent on the activity of the newly synthetized protein, it can be understandable that the evaluation of P-gp transport activity is of utmost importance. Overall, these results highlight that P-gp activity can be increased without an increase in P-gp expression.

Given the observed effects of the tested oxygenated xanthones on P-gp expression and activity, we further assessed the influence of those effects on the toxicity induced by PQ, a toxic P-gp substrate. For three of the tested compounds (OX2, OX4, and OX5), the observed modulatory effects on P-gp expression and activity caused a significant protection against PQ-induced cytotoxicity (with significant increases in the EC_50_ values of the PQ + OXs curves vs PQ only), showing their potential to protect against PQ intoxications by increasing its efflux. Noteworthy, xanthone OX3, although not interfering with P-gp expression or activity, significantly protected Caco-2 cells against the toxicity induced by the herbicide, which can be justified by other mechanisms involved, namely by the action of other carrier proteins. Furthermore, in the presence of Zos, a potent third-generation P-gp inhibitor, the observed decrease on the PQ-induced cell death was completely prevented. Despite the ability of the xanthones, OX1 and OX6, to significantly increase P-gp activity (24 h after exposure and immediately), they were not able to protect Caco-2 cells against the PQ-induced cytotoxicity.

Considering that in an intoxication scenario, the therapeutic action must be immediate, the oxygenated xanthone, OX6, was selected for the *ex vivo* studies, since it was demonstrated as the most promising direct activator of P-gp (augmented P-gp activity by 32%). *Ex vivo* methodologies are an experimental approach where an organ or tissue is removed from the animal and placed in chambers where physiological conditions found in the living body are mimicked (e.g., the access to nutrients and oxygen), allowing the viability of the organ or tissue during the experimental period. Furthermore, many studies use *ex vivo* approaches to accurately evaluate the ABC transporters’ function, especially P-gp [41,42,43]. For example, the modulatory effect of different compounds over ABC transporters expressed in intestinal epithelia can be assessed, *ex vivo*, by preparing everted intestinal sacs, which are then filled with a specific ABC transporter fluorescent substrate, and then the serosal to mucosal transport of the substrate is evaluated over time, in the presence or absence of the putative ABC transporter modulator [41,42,43]. Xanthone OX6 significantly increased the cumulative efflux of Rho 123 and this increase was blocked by Zos, proving P-gp involvement in Rho 123 efflux. The observed significantly higher serosal to mucosal transport of Rho 123 in OX6-treated everted sacs, an effect inhibited by Zos, reveals that P-gp expressed in the enterocytes of the rat ileum distal portion is activated by xanthone OX6. In fact, the increase in the transport of Rho 123 results from an immediate increase in its activity since a direct and a short-term contact between P-gp and the tested compound, OX6, occurred. Furthermore, and as mentioned, this effect in the pump activity was selectively blocked by Zos, a specific third-generation P-gp inhibitor, which proved the involvement of the protein in the efflux of the substrate. In accordance to the results obtained *in vitro*, it is possible to again conclude that xanthone OX6 behaves as a P-gp activator *ex vivo*, in the tested conditions, being able to immediately activate P-gp at the intestinal barrier, when a short contact between the drug and the everted intestinal sac occurs.

P-gp is an ATP binding cassette transporter formed by two TMDs (TMD1 and TMD2) that form the drug binding pockets and define the translocation pathway; and two cytoplasmic NBDs (NBD1 and NBD2) that bind and hydrolyse ATP, whose energy triggers drug transportation [33,34] (Figure 9). The number and location of the drug binding sites remain unclear. Initially, a large common binding site was assumed [44], but later a minimum of two binding sites was proposed to explain the complex behaviour of P-gp when cooperative, competitive, and noncompetitive interactions between MDR modulators were observed [45], which interacted in a positively cooperative manner [46]. P-gp was the first multidrug transporter for which structural data was obtained, albeit at low-to-medium resolution [47,48,49], which does not allow a structure-based design approach for the discovery of potential P-gp inhibitors. The effort to find new P-gp modulators has been accelerated after the crystal structure of several ABC transporters, such as Sav1866, became available [50]. However, mouse P-gp has a high sequence identity with human P-gp and, from a pharmacological perspective, is indistinguishable from the human isoform [51]. In 2009, the first X-ray crystal structure of a mammalian P-gp was resolved at 3.8 Å [51]. Therefore, the overall protein topology of the inward-facing conformation of P-gp, which represents an initial stage of the transport cycle that is competent for drug binding, has been established [51].

In 2013, two new X-ray structures of mouse P-gp in the apo form were reported [52] and in the following year, refined structures of mouse P-gp obtained by an electron density map generated from a single-wavelength anomalous dispersion phasing were described [53]. In 2015, four new X-ray structures of mouse P-gp at a 3.4 Å resolution provided a clearer understanding of the ligand entry, malleability of the binding, and induced helical movement of the transporter [35]. Therefore, this better resolution structure of mouse P-gp (pdb code 4Q9H) was selected for our molecular modelling study. Recently, the molecular structure of human ATP-bound P-glycoprotein in the outward-facing conformation, which has a lower affinity for substrates and modulators than the inward-facing conformation, was established by cryo-electron microscopy [54]. Homology modelling consists of building a protein model using a structural template, the template being a protein of known structure. The steps in homology modelling have been revised elsewhere [55]. The biopolymer module of Sybyl (distributed by Tripos) was used for the sequence alignment and model construction, as it has been described as a better performing software for building homology models [56].

The aim of this model construction was to obtain a 3D structure of human P-gp that could be used for the structure-based virtual screening of an in-house library of xanthones, in search of new potential P-gp modulators (inhibitors and activators of drug efflux). However, P-gp modulators’ and substrates’ promiscuity have long been a hallmark of P-gp activity. This transporter has a poly-specific drug-binding site, flanked by the TMD, as well as other sites for modulation, such as the ATP-binding site (on the NBD) and allosteric sites [57]. Thus, docking studies were performed on both the TMD and NBD. Controls for P-gp inhibition and activation were also used on the docking studies. The inhibitors used as controls are described as competitive inhibitors (first and second generation) by themselves being transported by P-gp and competing with other substrates [36], or noncompetitive inhibitors (third generation) that also bind to the drug-binding site, but lock the pump in a conformation that blocks drug efflux [58]. Therefore, verapamil, tamoxifen, and quinidine (1rst generation); dexniguldipine, biricodar, and sr33557 (2nd generation); and elacridar, tariquidar, and zosuquidar (3rd generation) were used as controls [36]. Flavonoids, such as flavonols, flavones, isoflavones, and chalcones [37], have been reported as MDR reversing agents by inhibiting the ATP-binding site of P-gp (located on the NBD) [59]. So, eight flavonoids (tangeretin, baicalein, nobiletin, sinensetin, quercetin, 2′,4′-dihydroxy-6′-methoxy-3′,5′-dimethylchalcone, 3,3′,4′,5,6,7,8-heptamethoxyflavone, and 3′,4′,7-trimethoxyflavone) and ATP were used as positive controls, as it is well documented that they bind with different affinity grades to the ATP-binding sites on NBD [36].

Also, P-gp activity can be directly increased by compounds that bind to P-gp and promote a conformational alteration that stimulates the transport of a substrate bound on another binding site [20,60], suggesting that the efflux pump contains several positively cooperative sites for drug binding and transport [46]. Therefore, leflunomide, blebbistatin, indirubin, and coelenteramide were used as controls for P-gp activation [38]. Docking results are represented on Table 3 and Figure 9.

Docking studies suggest that xanthone OX2, as well as other xanthonic derivatives that revealed low docking scores (such as xanthone OX3, OX4, OX5, and OX6) have potential for acting as P-gp modulators (induction and/or activation), although further investigation is needed to better understand the biological mechanism of action.

Further understanding of the potential location of the site of action of these molecules would require a systematic analysis of the ATPase profile [61,62] of combinations of verapamil and other transporter substrates with the new P-gp modulators, or even alanine or cysteine scanning mutagenesis [63,64].

Induction of P-gp has been regarded as one of the main mechanisms underlying anticancer drug-induced MDR. The induction mechanism of increasing P-gp expression without interfering with the protein transport cycle is another method that can be used to increase the efflux of cytotoxic compounds (apart from the previously mentioned activation mechanism) [7]. Although computational studies for the prediction of the modulation of P-glycoprotein are emerging [38], these do not address P-gp inducers as they have several potential mechanisms of action [5]. Due to the diversity of targets of P-gp inducers, the easiest way to address the problem of P-gp induction is through a ligand-based method [50]. Four pharmacophores for P-gp induction have already been described by our group [26]. The tested oxygenated xanthones were matched to previously described P-gp induction pharmacophores [26] and their fitting scores were evaluated (Table 4, pharmacophores I, III, and IV). Fitting with previously described pharmacophore for P-gp activation [2] has also been performed (Table 4, pharmacophore V) in order to infer which of the activities (induction and/or activation) is more relevant for protection against cytotoxic agents. The higher the fit value, the better is the mapping of the molecule to the pharmacophore. Accordingly, the xanthones, OX6 and OX3, and other xanthones have the potential of being P-gp inducers due to the high fitting scores obtained on P-gp induction pharmacophores (Figure 10A,B, Table 4, Pharmacophore I and IV). Compound OX4 is predicted as being the most active P-gp activator due to the high fitting score obtained on the P-gp activation pharmacophore (Figure 10C; Table 4, Pharmacophore V).

## 4. Materials and Methods

### 4.1. Materials

Rhodamine 123 (Rho 123), elacridar (Ela), zosuquidar (Zos), neutral red (NR) solution, and Dulbecco’s modified Eagle’s medium (DMEM) with 4.5 g/L glucose were obtained from Sigma (St. Louis, MO, USA). Reagents used in cell culture, such as nonessential amino acids (NEAA), heat-inactivated bovine serum (FBS), 0.25% trypsin/1 mM ethylenediamine tetraacetic acid (EDTA), antibiotic (10,000 U/mL penicillin, 10,000 μg/mL streptomycin), human transferrin (4 mg/mL), Hank’s balanced salt solution without calcium and magnesium (HBSS (-/-)), and phosphate-buffered saline solution (PBS), were purchased from Gibco Laboratories (Lenexa, KS). P-gp monoclonal antibody (clone UIC2) conjugated with PE was purchased from Abcam (Cambridge, United Kingdom). All the reagents used were of analytical grade or the highest grade available.

### 4.2. Synthesis of the Oxygenated Xanthones (OXs)

#### 4.2.1. General Information

All reagents and solvents were purchased from Sigma Aldrich (Sigma-Aldrich Co. Ltd., Gillinghan, UK) and no further purification process was implemented. Solvents were evaporated using a rotary evaporator under reduced pressure, Buchi Waterchath B-480. Microwave (MW) reactions were performed using an Ethos MicroSYNTH 1600 Microwave Labstation from Milestone (Thermo Unicam, Portugal). The internal reaction temperature was controlled by a fiber optic probe sensor. All reactions were monitored by thin-layer chromatography (TLC) carried out on precoated plates with 0.2 mm of thickness using Merck silica gel 60 (GF_254_) with appropriate mobile phases.

Flash column chromatography using silica gel 60 (0.040–0.063 mm, Merck, Darmstadt, Germany), flash cartridge chromatography (GraceResolv^®^, Grace Company, Deerfield, IL, USA), and Discovery^®^ DSC-SCX SPE cationic exchange cartridge (Grace Company, Deerfield, IL, USA) were used in the purification of the synthesized compounds. Melting points (m.p.) were measured in a Köfler microscope (Wagner and Munz, Munich, Germany) and are uncorrected. ^1^H- and ^13^C-NMR spectra were recorded at the University of Aveiro, Department of Chemistry in CDCl_3_ or DMSO-*d*_6_ (Deutero GmbH, Ely, UK) at room temperature on a Bruker Avance 300 spectrometer (300.13 MHz for ^1^H and 75.47 MHz for ^13^C, Bruker Biosciences Corporation, Billerica, MA, USA). Chemical shifts are expressed in δ (ppm) values relative to tetramethylsilane (TMS) as an internal reference. Coupling constants are reported in hertz (Hz). ^13^C-NMR assignments were made by bidimensional HSQC and HMBC NMR experiments (long-range C, H coupling constants were optimized to 7 Hz) or by comparison with the assignments of similar molecules. HRMS spectra were measured on a Bruker FTMS APEX III mass spectrometer (Bruker Corporation, Billerica, MA, USA) and recorded as electrospray ionization (ESI) mode in Centro de Apoio Cientifico e Tecnolóxico á Investigación (CACTI, University of Vigo, Pontevedra, Spain). The following compounds were synthesized and purified by the described procedures.

#### 4.2.2. Synthesis of 3,4-dimethoxy-1-methyl-9*H*-xanthen-9-one (OX1)

Xanthone OX1 (9.34 g, 63% yield) was synthesized from 1,2,3-trimethoxy-5-methylbenzene and characterized according to the previously described procedure (Figure 1) [23].

#### 4.2.3. Synthesis of 1-(dibromomethyl)-3,4-dimethoxy-9*H*-xanthen-9-one (OX2)

A mixture of xanthone OX1 (2.52 g, 9.32 mmol), *N*-bromosuccinimide (3.32 g, 18.7 mmol), and dibenzoyl peroxide (0.68 g, 2.8 mmol) in carbon tetrachloride (25 mL) was refluxed (85 °C) for 2 h. The reaction was monitored using *n*-hexane/ethyl acetate in a proportion of 8:2. Once completed the reaction, the resulting orange suspension was cooled at 0 °C and stirred for 30 min in an ice bath. The solid was filtered and washed with cold carbon tetrachloride. The filtrate was evaporated under reduced pressure and further purified by flash column chromatography (SiO_2_, *n*-hexane/ethyl acetate in gradient) to obtain the OX2 as a white powder.

1-(Dibromomethyl)-3,4-dimethoxy-9*H*-xanthen-9-one (OX2, Figure 1): White powder (3.14 g, 80% yield); ^1^H-NMR (CDCl_3_, 300.13 MHz): δ = 8.91 (1H, s, H-1), 8.30 (1H, dd, J = 8.0 and 1.6 Hz, H-8), 7.77 (1H, s, H-2), 7.73 (1H, ddd, J = 8.4, 7.0, and 1.6 Hz, H-6), 7.54 (1H, d, J = 8.4 Hz, H-5), 7.39 (1H, ddd, J = 7.5, H-7), 4.11 (3H, s, 4-OCH_3_), 4.05 (3H, s, 3-OCH_3_) ppm; ^13^C-NMR (CDCl_3_, 75.47 MHz): δ = 177.8 (C-9), 156.5 (C-3), 155.0 (C-10a), 150.4 (C-4a), 139.5 (C-4), 137.6 (C-1), 135.0 (C-6), 126.9 (C-8), 124.3 (C-7), 122.0 (C-8a), 117.6 (C-5), 111.9 (C-2), 110.8 (C-9a), 61.9 (3-OCH_3_), 56.6 (4-OCH_3_), 39.2 (C-1′) ppm. HRMS (ESI^+^): m/z [C_16_H_12_Br_2_O_5_ + H]^+^ calcd. for [C_16_H_13_Br_2_O_5_]: 426.91751; found: 426.91709.

#### 4.2.4. Synthesis of 3,6-dihydroxy-9-oxo-9H-xanthene-4,5-dicarbaldehyde (OX3)

Xanthone OX3 (280 mg, 4% yield) was synthesized from 3,6-dihydroxy-9*H*-xanthen-9-one and characterized according to the described procedure (Figure 1) [30].

#### 4.2.5. Synthesis of 4-hydroxy-3-methoxy-9-oxo-9H-xanthene-1-carbaldehyde (OX4)

Xanthone OX4 (661 mg, 35% yield) was synthesized from 4-hydroxy-3-methoxy-9*H*-xanthen-9-one and characterized according to the described procedure (Figure 1) [31].

#### 4.2.6. Synthesis of 3,4-dimethoxy-9-oxo-9H-xanthene-1-carbaldehyde (OX5)

Xanthone OX5 (152 mg, 80% yield) was synthesized from xanthone OX4 and characterized according to the described procedure (Figure 1) [25].

#### 4.2.7. Synthesis of 1-(hydroxymethyl)-3,4-dimethoxy-9H-xanthen-9-one (OX6)

Xanthone OX6 (21 mg, 52% yield) was synthesized from xanthone OX5 and characterized according to the described procedure (Figure 1) [25].

### 4.3. Caco-2 Cell Culture

Human colorectal adenocarcinoma cells (Caco-2 cells) were obtained from the American Type Culture Collection (ATTCC; Manassas, VA, USA). Caco-2 cells were cultured in 75 cm^2^ flasks (T75) using Dulbecco’s modified Eagle’s medium (DMEM) supplemented with 10% fetal bovine serum (FBS), 100.0 µM nonessential amino acids (NEAA), 100.0 U/mL penicillin, and 100.0 µg/mL streptomycin. Cells were sustained in a 5% CO_2_-95% air atmosphere at 37 °C, and the medium was changed every 2 days. Cultures were passaged weekly by trypsinization (0.25% trypsin/1.0 mM EDTA). In all experiments, the cells were seeded at a density of 60,000 cells/cm^2^, and used 4 days after seeding, when confluence was reached.

A 50.0 mM stock solution of each OX was prepared in DMSO. All stock solutions were stored at −20 °C and freshly diluted on the day of the experiment in cell culture medium (ensuring that DMSO did not exceed 0.04% of the exposure media, except in the cytotoxicity assays where the 50.0 μM OXs concentration corresponds to a 0.1% DMSO concentration).

### 4.4. Oxygenated Xanthones’ Cytotoxicity

Oxygenated xanthones (OXs 1–6, 0–50.0 µM) cytotoxicity was evaluated by the neutral red (NR) uptake assay, 24 h after exposure.

#### 4.4.1. Neutral Red Uptake Assay

The NR uptake assay is used to estimate the cell cytotoxicity or viability and it is based on the ability of viable cells to incorporate and bind the supravital dye neutral red in the lysosomes [65]. Cells were seeded onto 96-well tissue culture plates at a density of 60,000 cells/cm^2^ and, when confluence was reached, the cells were exposed to the six xanthones (0–50.0 µM) in fresh cell culture medium. Twenty-four hours after the exposure, the cells were incubated with NR (50.0 µg/mL in cell culture medium), at 37 °C in a 5% CO_2_-95% humidified air atmosphere, for 60 min. At the end of the incubation period, the cell culture medium was removed, the dye absorbed only by viable cells extracted with absolute ethyl alcohol/distilled water (1:1) with 5% acetic acid, and the absorbance measured at 540 nm in a multiwell plate reader (PowerWaveX BioTek Instruments, Vermont, VT, USA). Cytotoxicity was evaluated by the percentage of NR uptake relative to the control cells (0 μM). Four independent experiments were performed, in triplicate.

### 4.5. Evaluation of P-Glycoprotein Expression

Caco-2 cells were seeded onto 12-well plates, at a density of 60,000 cells/cm^2^ and, four days after seeding, when confluence was reached, the cells were exposed to the tested xanthones, OX 1–6, at a noncytotoxic concentration (20.0 µM) prepared in fresh cell culture medium. Twenty-four hours after exposure, the cells were washed twice with PBS buffer (pH 7.4) and harvested by trypsinization (0.25% trypsin/1 Mm EDTA) to obtain a cell suspension. This cell suspension was centrifuged (300 g, for 5 min), and suspended in PBS buffer (pH 7.4) containing 10% heat-inactivated FBS and the P-gp antibody (UIC2 clone conjugated with PE). After 45 min of incubation at 37 °C, in the dark, and under gentle shaking, the cells were washed twice with PBS buffer (pH 7.4) containing 10% heat-inactivated FBS, centrifuged (300 g for 5 min), and kept on ice until the analysis. The cells were suspended on ice-cold PBS buffer immediately before the cytometer analysis. Cells were analysed in a BD Accuri^TM^ C6 flow cytometer (BD Biosciences, CA, USA).

The antibody used in this experiment was diluted according to the manufacturer’s instructions for flow cytometry. To estimate the non-specific binding of the anti-P-gp antibody (UIC2), mouse IgG2a was used as an isotype-matched negative control.

The fluorescence of the UIC2-PE-antibody was measured by a 585 ± 40 nm band-pass filter (FL2 detector). The logarithmic fluorescence was recorded and displayed as a single parameter histogram and based on the acquisition of data for at least 20,000 cells. The parameter used for comparison was the mean of the fluorescence intensity (MFI), calculated as a percentage of the control (0 μM OXs). To eliminate the possible contribution of cells’ autofluorescence to the analysed fluorescence signals, unlabelled cells (with or without being exposed to the tested OXs) were also analysed in every experiment. Representative histograms of the flow cytometry analysis of Caco-2 cells’ autofluorescence in the 585 ± 40 nm band-pass filter (FL2 detector), 24 h after the incubation with the tested oxygenated xanthones, OX 1–6 (20.0 μM), are illustrated in Appendix A. At least five independent experiments were performed in duplicate.

### 4.6. Evaluation of P-Glycoprotein Transport Activity

P-gp transport activity was evaluated by flow cytometry using Rho 123 (2.0 µM) as a P-gp fluorescent substrate. Accordingly, two different protocols were performed: Rho 123 accumulation assay in the presence of the tested OXs only during the accumulation of the fluorescence substrate, and Rho 123 accumulation assay in cells pre-exposed to the OXs for 24 h. To eliminate the possible contribution of cells’ autofluorescence to the analysed fluorescence signals, unlabelled cells (with or without being exposed to the tested OXs) were also analysed in every experiment. Representative histograms of the flow cytometry analysis of Caco-2 cells’ autofluorescence in the 530 ± 15 nm band-pass filter (FL1 detector), 24 h after the incubation with the tested oxygenated xanthones, OX 1–6 (20.0 μM), are illustrated in Appendix A.

#### 4.6.1. Rhodamine 123 Efflux in the Presence of the Tested Oxygenated Xanthones

Caco-2 cells were seeded onto 75 cm^2^ flasks and, after reaching confluence, washed with HBSS (-/-) and harvested by trypsinization (0.25% trypsin/1 mM EDTA). The cell suspension was then divided into several aliquots of 300,000 cells/mL. Thereafter, the cells were centrifuged (300 g for 10 min), suspended in HBSS (-/-) containing 10% heat-inactivated FBS and 2.0 µM μM Rho 123, with or without simultaneous exposure to tested xanthones (20.0 μM OX 1–6), and incubated at 37 °C in a shaking water bath for 60 min, in the presence (IA) or in the absence (NA) of the P-gp inhibitor (Elacridar (Ela), 10.0 μM). After this accumulation period, the cells were washed twice with ice-cold HBSS (-/-) with 10 % heat-inactivated FBS, centrifuged (300 g for 10 min) at 4° C, and kept on ice until the flow cytometry analysis.

The fluorescence measurements of isolated cells were achieved as reported in Section 4.5. *Evaluation of P-Glycoprotein Expression*. Rho 123 fluorescence was measured by a 530 ± 15 nm band-pass filter (FL1 detector) and P-gp activity was evaluated by the Rho 123 accumulation ratio (Equation (1)), and expressed as a percentage of the control (0 μM OXs).
(1)$$\mathrm{Rho}\;123\;\mathrm{accumuation}=\frac{\mathrm{MFI}\;\mathrm{of}\;\mathrm{Rho}\;123\;\mathrm{accumuation}\;\mathrm{under}\;\mathrm{inhibition}\;\left(\mathrm{IA}\right)}{\mathrm{MIF}\;\mathrm{of}\;\mathrm{Rho}\;123\;\mathrm{normal}\;\mathrm{accumuation}\;\left(\mathrm{NA}\right)}$$

Equation (1) P-glycoprotein (P-gp) activity assessed by the ratio between the amount of Rhodamine 123 (Rho 123) accumulated under inhibition conditions (Elacridar (Ela), 10.0 μM) and the amount of accumulated Rho 123 in the absence of the P-gp inhibitor.

With the increase in P-gp activity, the amount of Rho 123 effluxed by this pump is also increased, which is accompanied by a decrease in the intracellular fluorescence intensity as a consequence of the decrease in the Rho 123 intracellular content. Therefore, a higher MFI_IA_/MFI_NA_ ratio results from a lower value of MFI_NA_, which is due to an increase in P-gp activity, since Rho 123 is being pumped out of the cells during the accumulation phase in the absence of the inhibitor. Six independent experiments were performed in triplicate.

#### 4.6.2. Rhodamine 123 Efflux in Cells Pre-Exposed to the Tested Oxygenated Xanthones

For the P-gp activity evaluation 24 h after the xanthones’ exposure, Caco-2 cells were seeded in 12-well plates at a density of 60,000 cells/cm^2^. Four days after seeding, when confluence was reached, the cells were exposed to the tested oxygenated xanthones (OX 1–6, 20.0 μM), in fresh cell culture medium during a 24 h period. After the incubation period, the cells were washed with HBSS (-/-) and harvested by trypsinization (0.25% trypsin/1 mM EDTA) to obtain a cell suspension. Subsequently, the cells from each well were divided in two aliquots: The first aliquot was submitted to a Rho 123 accumulation phase performed under inhibited conditions (Rho 123 accumulation in Ela presence, IA); and the second aliquot was submitted to a Rho 123 accumulation phase achieved under normal conditions (Rho 123 accumulation in Ela absence, NA).

In IA accumulation, the cells were centrifuged (300 g for 10 min), suspended in HBSS (-/-) containing 10% heat-inactivated FBS, 2.0 µM Rho 123, and Ela (10.0 µM), and incubated for 60 min at 37 °C in a water bath with slight shaking and protected from the light, allowing the maximum accumulation of the fluorescent substrate, since P-gp was inhibited. In its turn, for NA accumulation, an analogous incubation was performed, but in the absence of Ela. For both aliquots, after the incubation period, the cells were washed twice with ice-cold HBSS (-/-) containing 10% heat-inactivated FBS and suspended in an ice-cold wash solution immediately before analysis.

Flow cytometry analysis was performed according to the previously described (“Section 4.6.1. Rhodamine 123 efflux in the presence of the tested oxygenated xanthones”), with the green intracellular fluorescence of Rho 123 measured using a 530 ± 15 nm band-pass filter (FL1 detector). The results were obtained according to Equation (1) and expressed as a percentage of the control (0 μM OXs). Six independent experiments were performed in duplicate.

### 4.7. Paraquat Cytotoxic Assays

Paraquat (PQ) cytotoxicity was evaluated in Caco-2 cells by the NR uptake assay, with and without simultaneous incubation with the xanthonic derivatives. Briefly, the cells were seeded onto 96-well plates and exposed, four days after seeding (when confluence was reached), to 0–5000 μM PQ in the presence and in the absence of tested xanthones, at a noncytotoxic concentration (20.0 μM). Twenty-four hours after exposure, PQ cytotoxicity was evaluated by the NR uptake assay as previously described in the “Section 4.4.1. Neutral red uptake assay” section. Four independent experiments were performed in triplicate.

To verify the involvement of P-gp in the OXs’ protective effects, these procedures were repeated in the presence of a potent P-gp inhibitor (Ela, 10.0 μM). Four independent experiments were performed in triplicate. The inhibitor cytotoxicity was previously assessed by the neutral red uptake assay 24 h after exposure, as performed in the evaluation of xanthones’ cytotoxicity.

### 4.8. Ex Vivo Studies

#### 4.8.1. Animals

Male Wistar-Han rats (238 g–316 g) were obtained from the Rodent Animal House Facility, from the Abel Salazar Biomedical Sciences Institute, University of Porto (Portugal), and were kept under standard laboratory conditions (12/12 h light/dark cycles, 22 ± 2 °C room temperature, 50%–60% humidity) and had free access to water and pellet food. Before the experiments, animals were fasted for 12 h and water containing 1% sugar was provided ad libitum. Animal experiments were approved by the Organismo Responsável pelo Bem-Estar Animal (ORBEA; protocol number 250/2018) from the Abel Salazar Biomedical Sciences Institute and submitted for evaluation and licensing to the Portuguese General Directorate of Veterinary Medicine. Housing and experimental treatment of the animals were in accordance with the guidelines defined by the European Council Directive (2010/63/EU).

#### 4.8.2. Effect of Oxygenated Xanthone OX6 on P-Glycoprotein Activity–*Ex Vivo*

To evaluate the effect of xanthone OX6 on P-gp activity, rats were anesthetized intraperitoneally (i.p.) with ketamine/xilazine at 270 and 30 mg/kg body weight (b.w.) doses, respectively, and the distal portion of the ileum (20 cm) was removed, gently rinsed with ice-cold saline solution, and immediately used to prepare the everted intestinal sacs. The intestine portions were carefully everted and each sac (approximately 10 cm length) was placed in a 5 mL KH-containing chamber (40.0 mM glucose, pH 7.4), with or without the addition of xanthone OX6 (20.0 μM), and in the presence or absence of Zos (10.0 μM), a known third-generation P-gp inhibitor. The mucosal medium was continuously gassed with 95% O_2_–5% CO_2_ and maintained at 37 °C. After a 10 min stabilization period, the serosal compartment was filled with 1 mL RHO 123 (300 μM) prepared in KH buffer, used as a model P-gp substrate. Aliquots of 100 μL of mucosal medium were sampled every 5 min for a 45-min period. Then, the sacs were emptied and gently dried and weighed. Rho 123 concentration was determined spectrofluorometrically in all samples taken from the mucosal medium (in a multi-well plate reader at excitation/emission wavelengths of 485/528 nm, respectively), using a Rho 123 calibration curve prepared in KH buffer. The amount of Rho 123 was then normalized per milligram of tissue [42].

### 4.9. Statistical Analysis

GraphPad Prism 6 for Windows (GraphPad Software, San Diego, CA, USA) was used to perform all statistical calculations. Three tests were performed to check the normality of the data distribution: Kolmogorov-Smirnov, D’Agostino & Pearson omnibus, and Shapiro-Wilk normality tests. For data with a parametric distribution, one-way ANOVA was used to do the statistical comparisons, followed by the Dunnett’s multiple comparisons test or the Dunn’s post hoc test. For data with only two groups (e.g., elacridar cytotoxicity), the statistical comparisons were made using the unpaired t test. Statistical comparisons between groups in experiments with two variables (PQ cytotoxicity assays and *ex vivo* studies) were made using two-way ANOVA, followed by the Sidak’s multiple comparison post hoc test or by the Tukey’s multiple comparisons test. In the PQ cytotoxicity assays, concentration—response curves were fitted using least squares as the fitting method and the comparisons between curves (LOG EC_50_, TOP, BOTTOM, and Hill Slope) were made using the extra sum-of-squares F test. Each figure has a legend with all the performed statistical analysis. In all cases, *p* values lower than 0.05 were considered significant.

### 4.10. In Silico Studies

#### 4.10.1. P-Glycoprotein Model Construction

All studies were performed on a Sony Vaio Intel^®^ Core™i5 CPU with a 2.67 GHz processor using the Windows 7 operating system. Human P-gp target sequence was obtained in FASTA format. A psi-blast search was run to produce an initial sequence alignment, which served as input for the sequence—structure homology recognition algorithm, FUGUE [66], implemented in Sybyl (Tripos Certara, MO, USA) [67], which identified structural homolog families within the HOMSTRAD database [68]. Target 3D structure was chosen based on % identity and the availability of 3D coordinates with the best resolution: 4Q9H mice P-gp, with a % identity of 89.0%, and a resolution of 3.4 Å [35].

The single sequence alignments between the target and 4Q9H obtained from FUGUE was submitted to Orchestrar [69], which aligned the homolog structurally and then identified conserved regions using a clustering process (SCR). Choral constructs a 3D model of the target sequence using the SCRs as a template [70]. Bridge, Fread, and Petra programs were used for the loop search [71]. The top ranked loops found for segments Glu24:Lys31, and Ser1272:Gln1280, based on the energy of the predicted fragment in the context of its placement in the model (Ek), were added to the model using the Tuner program. Andante was used to add side chains to the backbone model by borrowing side chains’ dihedral information from homologues while avoiding steric clashes [72]. In addition, Biopolymer was used to examine the overall geometry of the models, including the peptide bond distances and torsions [73]. Hydrogen atoms were added and the terminal residues were maintained in their charged form. All the amide moieties in the side chains of asparagine and glutamine were adjusted to optimize their interactions with surrounding residues and groups of atoms. Charges were added using MMFF94. The model was subjected to a staged minimization consisting of the minimization of hydrogen positions, side chain atoms, and the backbone. In each step, 5000 cycles of the steepest descent and conjugate gradient energy minimizations were carried out using MMFF94. Model validation was done by a single-point energy calculation and Ramachandran plot analysis [74].

#### 4.10.2. Docking Study

The 3D structures of the six tested oxygenated xanthones and controls were drawn using HyperChem 7.5 [75], being minimized by the semi-empirical Polak–Ribiere conjugate gradient method (RMS < 0.1 kcal/Å mol) [76]. Docking simulations between the P-gp model and the small molecules were undertaken in AutoDock Vina (Scripps Research Institute, USA) [77,78]. AutoDock Vina considered the target conformation as a rigid unit while the ligands were allowed to be flexible and adaptable to the target. Vina searched for the lowest binding affinity conformations and returned nine different conformations for each ligand. AutoDock Vina was run using an exhaustiveness of 8 and a grid box with the dimensions of 37.0, 30.0, and 40.0 Å, engulfing the channel formed by the transmembrane domains; or 20.0, 30.0 and 30.0, 32.0, and 25.0 Å, engulfing the ATP-binding site on NBD. Conformations and interactions were visualized using PyMOL version 1.3 [79]. Two validation procedures were performed: Enrichment analysis (enrichment factor (EF) and receiver operating characteristic (ROC) curve), and test set prediction.

#### 4.10.3. Mapping of Small Molecules onto Pharmacophores

The mapping of small molecules onto previously described P-gp induction pharmacophores [26] and P-gp activation pharmacophores [2] was performed. *In silico* studies were accomplished using the “Best Fit” method in catalyst (Accelrys 2.1, San Diego, CA, USA). During the flexible fitting process, conformations of the tested molecules were calculated within the 20 kcal/mol energy threshold. Fitting was evaluated by the analysis of the fit score.

## 5. Conclusions

In conclusion, the tested oxygenated xanthones demonstrated, at the intestinal level, the capacity for P-gp modulation, opening new perspectives to mechanistically explore the physiological role of this transporter. These compounds also showed the potential to protect Caco-2 cells against the toxicity of the herbicide, PQ. In addition, xanthone OX6 confirmed the *in vitro* results, by significantly increasing the efflux of Rho 123 *ex vivo*, in rat intestinal everted sacs. Docking simulations predicted these derivatives to bind at the same drug-binding pocket as the known modulators. Pharmacophoric mapping suggests that the xanthones may be acting as activators and/or inducers of P-gp. Moreover, xanthones deserve attention at the toxicological level, as they can constitute a promising new source of novel derivatives that are worthy of further study.

## Figures and Tables

**Figure 1 molecules-24-00707-f001:**
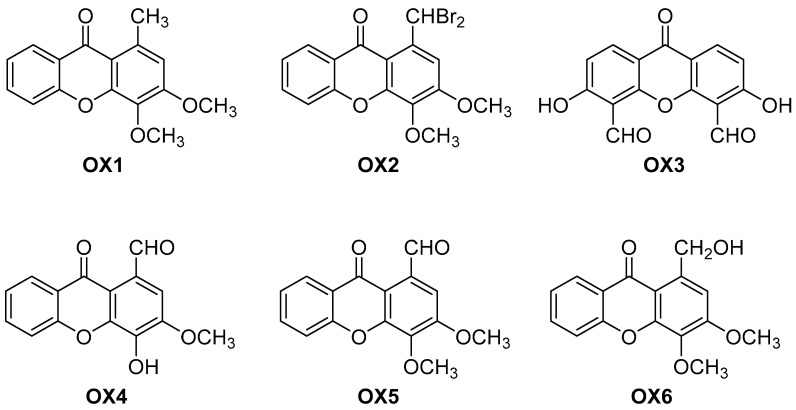
Chemical structures of the oxygenated xanthones OX 1–6 investigated in this study.

**Figure 2 molecules-24-00707-f002:**
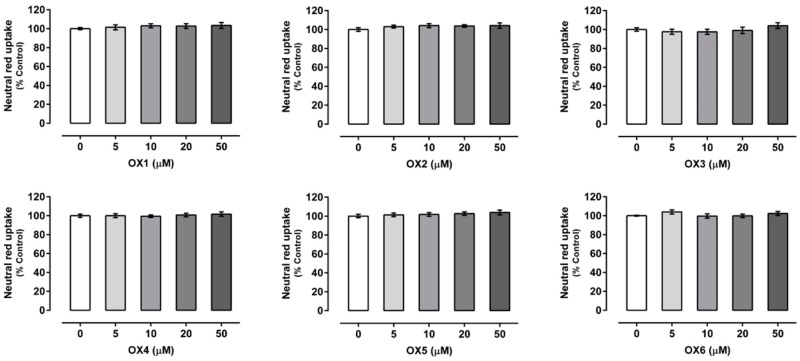
Oxygenated xanthones (OX 1–6, 0–50.0 µM) cytotoxicity in Caco-2 cells evaluated by the neutral red (NR) uptake assay, 24 h after incubation. Results are presented as mean ± SEM from four independent experiments, performed in triplicate. Statistical comparisons were made using the parametric method of one-way ANOVA, followed by the Dunnett’s multiple comparisons test.

**Figure 3 molecules-24-00707-f003:**
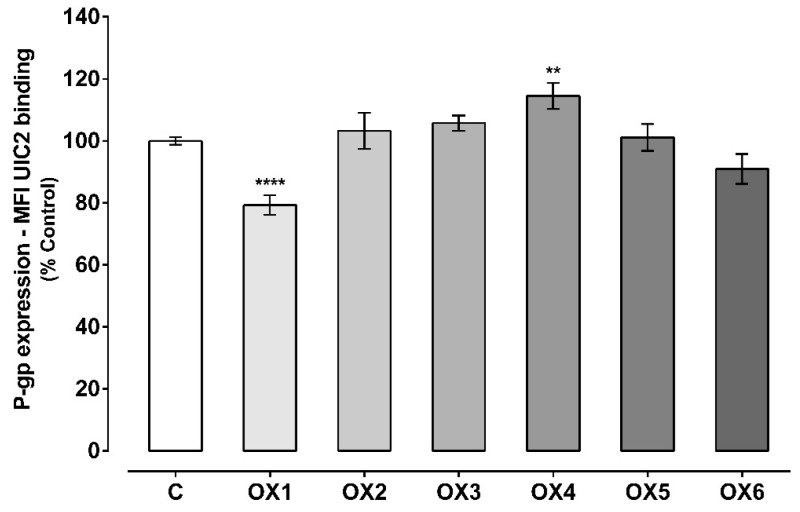
P-glycoprotein (P-gp) expression levels in Caco-2 cells exposed to the six tested oxygenated xanthones (OX 1–6, 20.0 μM) for 24 h. Results are presented as mean ± SEM from five independent experiments (performed in duplicate). Statistical comparisons were made using one-way ANOVA, followed by Dunnett’s multiple comparisons test, [** *p* < 0.01; **** *p* < 0.0001 vs. control (0 μM)].

**Figure 4 molecules-24-00707-f004:**
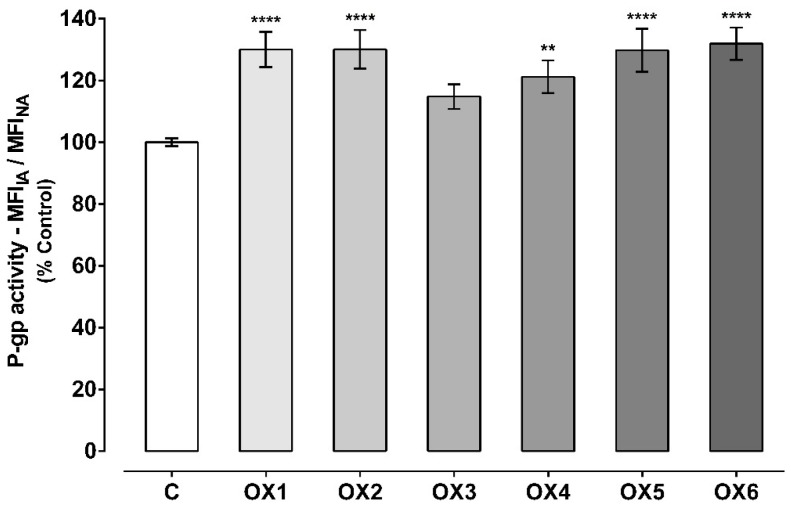
P-glycoprotein (P-gp) activity assessed through the Rhodamine 123 (Rho 123) accumulation in the presence of the six tested oxygenated xanthones (OX 1–6, 20.0 µM), during the Rho 123 accumulation phase. Results are presented as mean ± SEM from six independent experiments (performed in triplicate). Statistical comparisons were made using one-way ANOVA, followed by Dunnett’s multiple comparisons test [** *p* < 0.01; **** *p* < 0.0001 vs. control (0 μM)].

**Figure 5 molecules-24-00707-f005:**
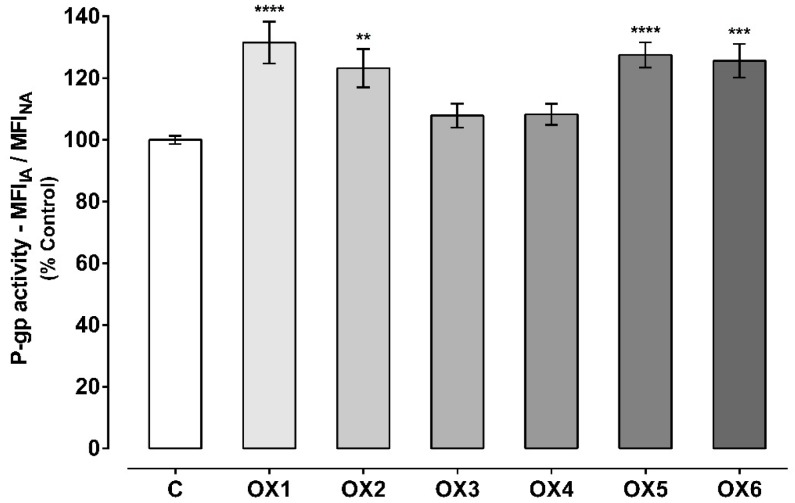
P-glycoprotein (P-gp) activity evaluated through the Rhodamine 123 (Rho 123) accumulation assay in Caco-2 cells pre-exposed to the six tested oxygenated xanthones (OX 1–6, 20.0 µM) for 24 h. Results are presented as mean ± SEM from six independent experiments (performed in duplicate). Statistical comparisons were estimated using one-way ANOVA, followed by Dunnett’s multiple comparisons test [** *p* < 0.01; *** *p* < 0.001; **** *p* < 0.0001 vs. control (0 μM)].

**Figure 6 molecules-24-00707-f006:**
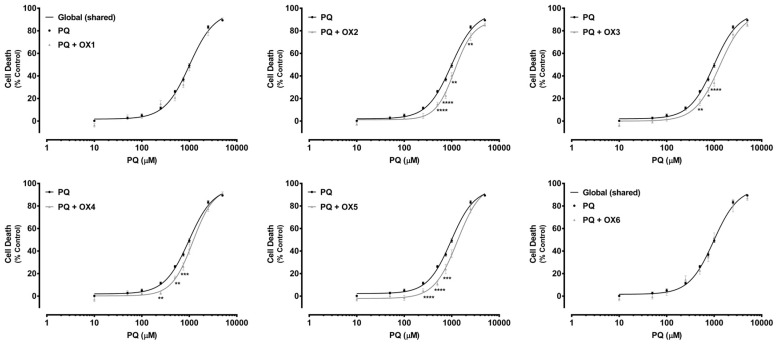
Paraquat (PQ) concentration—response (cell death) curves obtained in the absence (PQ curve) or in the presence of the six tested oxygenated xanthones (20.0 μM) (PQ + OX 1–6 curves). Results are presented as mean ± SEM from four independent experiments (performed in triplicate). Concentration—response curves were fitted using least squares as the fitting method and the comparisons between PQ and PQ + OX 1–6 curves (LOG EC_50_, TOP, BOTTOM, and Hill Slope) were made using the extra sum-of-squares F test. Statistical comparisons were made using two-way ANOVA, followed by the Sidak’s multiple comparisons post hoc test (* *p* < 0.05; ** *p* < 0.01; *** *p* < 0.001; **** *p* < 0.0001 PQ + OXs vs. PQ). In all cases, *p* values < 0.05 were considered statistically significant.

**Figure 7 molecules-24-00707-f007:**
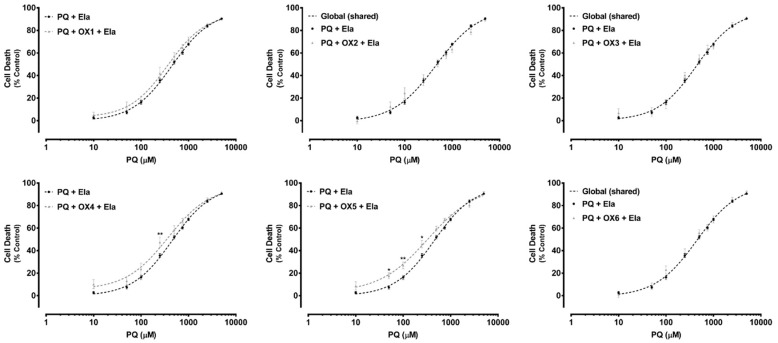
Paraquat (PQ) concentration—response (cell death) curves obtained in the presence of Elacridar (Ela, 10.0 µM) with (PQ + OXs + Ela curves) and without (PQ + Ela curve) exposure to the six tested oxygenated xanthones, OX 1–6 (20.0 µM). Results are presented as mean ± SEM from four independent experiments (performed in triplicate). Concentration—response curves were fitted using least squares as the fitting method and the comparisons between PQ + Ela and PQ + OXs + Ela curves (LOG EC_50_, TOP, BOTTOM, and Hill Slope) were made using the extra sum-of-squares F test. Statistical comparisons were made using two-way ANOVA, followed by the Sidak’s multiple comparisons post hoc test (* *p* < 0.05; ** *p* < 0.01 PQ + OX 1-6 + Ela vs. PQ + Ela). In all cases, *p* values < 0.05 were considered statistically significant.

**Figure 8 molecules-24-00707-f008:**
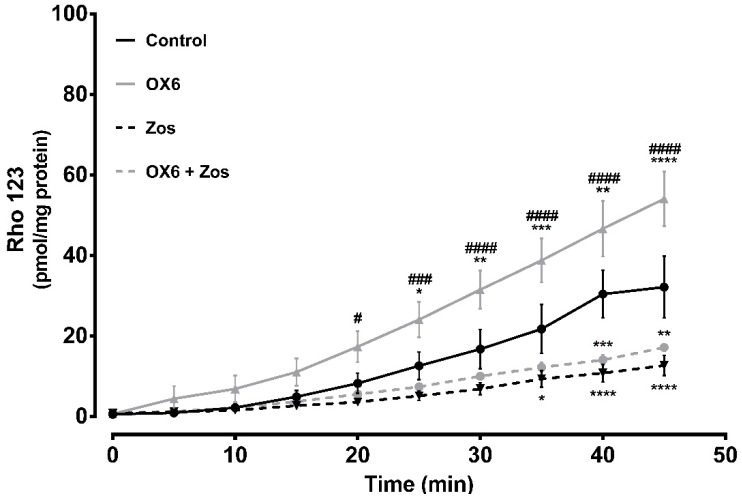
Short-term and direct effect of xanthone OX6 on P-glycoprotein (P-gp) activity, evaluated *ex vivo*. P-gp-mediated Rhodamine 123 (Rho 123) efflux was measured using 10 cm everted sacs from distal ileum. The sacs were filled with 300.0 μM Rho 123 (serosal side). The dye secreted into the outside compartment (mucosal side) was assessed every 5 min for 45 min, in the presence or absence of 20.0 μM OX6 and/or 10.0 μM zosuquidar (Zos). Data are expressed as means ± SEM of five to eight rats per group. The excreted amounts of Rho 123 into the mucosal side were evaluated by spectrofluorometry, using a Rho 123 calibration curve, and expressed as pmol of Rho 123 transported per mg of tissue. Statistical comparisons were made using two-way ANOVA followed by the Tukey’s multiple comparisons post hoc test (* *p* <0.05; ** *p* <0.01; *** *p* < 0.001; **** *p* < 0.0001 vs. control; ^#^
*p* < 0.05; ^###^
*p* < 0.001; ^####^
*p* < 0.0001 OX6 vs. OX6 + Zos).

**Figure 9 molecules-24-00707-f009:**
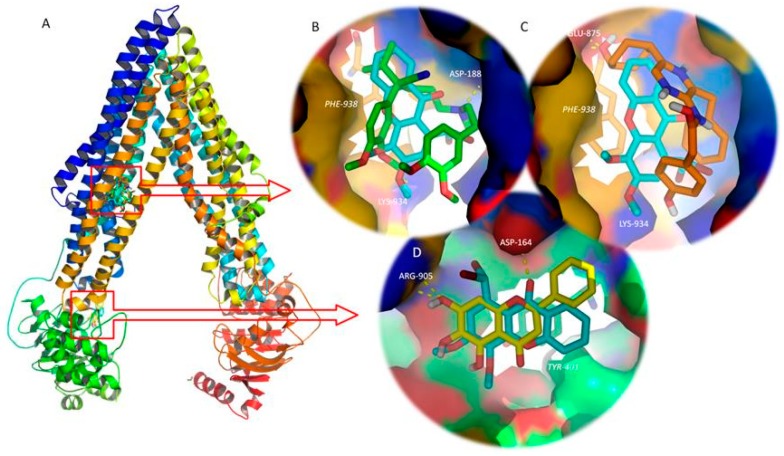
(**A**) Ribbon representation for the P-glycoprotein (P-gp) model and control/tested molecules docked to the transmembrane domains (TMD, top) and nucleotide binding domain (NBD, bottom). (**B**) Detailed view of the P-gp inhibitor, verapamil (green sticks), and tested xanthone OX2 (blue sticks) on the drug binding site on the interface of the TMDs. (**C**) Detailed view of the P-gp activator, coelenteramide (orange sticks), and tested xanthone OX2 (blue sticks) on the drug binding site on the interface of the TMDs. (**D**) Detailed view of the P-gp inhibitor, baicalein (yellow sticks), and test xanthone OX2 (blue sticks) on the NBD. Polar interactions are represented as yellow broken lines and evolved residues are labelled. Residues involved in stacking interactions are represented in sticks and labelled in italics.

**Figure 10 molecules-24-00707-f010:**
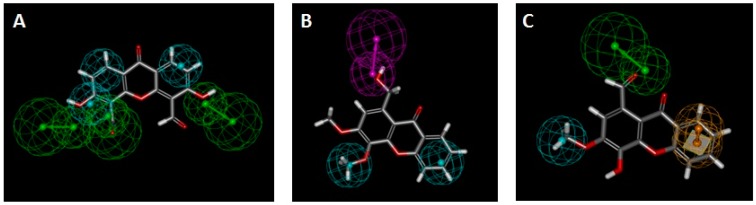
(**A**) Xanthone OX3 fit to pharmacophore I (P-gp induction); (**B**) Xanthone OX6 fit to pharmacophore IV (P-gp induction); (**C**) Xanthone OX4 fit to pharmacophore V (P-gp activation). Green spheres = hydrogen acceptor; pink spheres = hydrogen-donor; blue sphere = hydrophobic; yellow spheres = aromatic.

**Table 1 molecules-24-00707-t001:** EC_50_ (half-maximum-effect concentrations), TOP (maximal effect), BOTTOM (baseline), and Hill Slope values of the paraquat (PQ) concentration—response curves, with (PQ + OXs) or without (PQ) simultaneous exposure to the six tested oxygenated xanthones, OX 1–6 (20.0 µM).

	PQ	PQ + OX1	PQ + OX2	PQ + OX3	PQ + OX4	PQ + OX5	PQ + OX6
EC_50_ [half-maximum-effect concentrations, µM (95% CI)]	982.4 (912.4–1058)	1124 (900.8–1402)	**1153 *** (1013–1313)	**1291 **** (1050–1587)	**1188 *** (1027–1375)	**1217 **** (1046–1417)	943.0 (757.2–1174)
Top (maximal cell death, % control)	96.65	~100.0	88.76	96.80	96.48	94.89	94.51
Bottom (baseline, % control)	2.008	0.9805	1.236	0.01597	0.2445	−1.129	−0.2018
Hill slope	1.694	1.524	2.178	1.674	1.955	2.088	1.618
Curve *p* value (Comparison between the fitted curves)	-	0.2139	**<0.0001**	**<0.0001**	**<0.0001**	**<0.0001**	0.6121

Concentration—response curves were fitted using least squares as the fitting method and the comparisons between PQ and PQ + OX 1-6 curves were made using the extra sum-of-squares F test. In all cases, *p* values < 0.05 were considered significant (* *p* < 0.05; ** *p* < 0.01 for PQ vs. PQ + OXs). 95% CI–95% Confidence Intervals. Bold is used when significant exists.

**Table 2 molecules-24-00707-t002:** EC_50_ (half-maximum-effect concentrations), TOP (maximal effect), BOTTOM (baseline), and Hill Slope values of the paraquat (PQ) concentration—response curves, in the presence of the P-gp inhibitor, Elacridar (Ela, 10.0 µM), with (PQ + OXs + Ela) or without (PQ + Ela) simultaneous exposure to the six tested oxygenated xanthones, OX 1–6 (20.0 µM).

	PQ + Ela	PQ + OX1 + Ela	PQ + OX2 + Ela	PQ + OX3 + Ela	PQ + OX4 + Ela	PQ + OX5 + Ela	PQ + OX6 + Ela
EC_50_ [half-maximum-effect concentrations, µM (95% CI)]	450.3 (397.4–510.4)	362.5 (278.8–471.4)	430.6 (403.5–525.4)	418.6 (324.3–540.3)	369.2 (260.3–523.5)	334.7 (243.5–460.0)	397.8 (293.0–540.0)
Top (maximal cell death, % control)	97.76	94.80	~100.0	96.10	97.37	97.84	~100.0
Bottom (baseline, % control)	−0.2621	2.862	−4.114	2.252	5.335	2.986	−2.547
Hill slope	1.039	1.086	0.7837	1.105	0.9732	0.8449	0.8964
Curve *p* value (Comparison between the fitted curves)	-	**0.0092**	0.0591	0.5798	**<0.0001**	**<0.0001**	0.2742

Concentration—response curves were fitted using least squares as the fitting method and the comparisons between PQ + Ela and PQ + OX 1–6 + Ela curves were made using the extra sum-of-squares F test. In all cases, *p* values < 0.05 were considered significant. 95% CI—95% Confidence Intervals. Bold is used when significant exists.

**Table 3 molecules-24-00707-t003:** Docking scores for the six tested oxygenated xanthones (OX 1–6) and controls on P-glycoprotein (P-gp) transmembrane domains (TMD, A) and nucleotide binding domains (NBD, B).

	A			B	
	Ligand	Free Energy of Ligand: P-gp TMD Complex (Kcal.mol^−2^)		Ligand	Free Energy of Ligand: P-gp NBD Complex (Kcal.mol^−2^)
	OX1	−6.5		OX1	−6
	OX2	−7		OX2	−6.2
	OX3	−6.7		OX3	−5.9
	OX4	−6.7		OX4	−5.8
	OX5	−6.7		OX5	−5.8
	OX6	−6.7		OX6	−6.1
P-gp TMD inhibitors (controls)	Verapamil	−6	P-gp NBD inhibitors (controls)	Tangeretin	−6
	Tamoxifen	−7.3		Baicalein	−6.2
	Quinidine	−7.7		Nobiletin	−5.8
	Dexniguldipine	−7.2		Sinensetin	−5.7
	Biricodar	−7		Quercetin	−5.9
	SR33557	−6.5		2′,4′-Dihydroxy-6′-methoxy-3′,5′-dimethylchalcone	−6.1
	Elacridar	−9.1		3,3′,4′,5,6,7,8-Heptamethoxyflavone	−6
	Zosuquidar	−8.6		3′,4′,7-Trimethoxyflavone	−5.9
	Tariquidar	−8.4			
P-gp activators (controls)	Indirubin	−7.6			
	Coelenteramide	−8.7			
	Blebbistatin	−8.8			
	Lefluromide	−7.7			

**Table 4 molecules-24-00707-t004:** Fit values of the six tested oxygenated xanthones OX 1–6 to P-glycoprotein (P-gp) induction pharmacophores I, III, and IV [26] and to P-gp activation pharmacophore V [2].

	Fit Values			
Compound	Pharmacophore I (6 features) 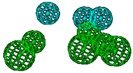	Pharmacophore III (5 features) 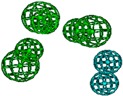	Pharmacophore IV (3 features) 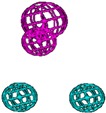	Pharmacophore V (3 features) 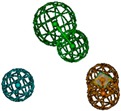
OX1	0.485713	No match	1.99999	2.40877
OX2	1.01167	No match	2	2.91681
OX3	2.14728	1.77954	1.98371	2.98381
OX4	0.697596	1.7693	1.84491	2.99583
OX5	1.2425	1.50716	1.99911	2.81259
OX6	1.37048	1.84221	2.91711	2.62604

Green spheres = hydrogen acceptor; pink spheres = hydrogen-donor; blue sphere = hydrophobic; yellow spheres = aromatic.

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
