# Peer review of "Newly Synthesized Oxygenated Xanthones as Potential P-Glycoprotein Activators: In Vitro, Ex Vivo, and In Silico Studies"

_molecules, 2019, doi:10.3390/molecules24040707_

Round 1

Reviewer 1 Report

Paper have to be rewritten according following recommendations prior its acceptance

1.       Cytometric observation in figures 3-5 have to be documented also by FACS histograms (not only by column plots) directly  in paper or at  last in supplementary data

2.       Alternation of P-gp expression have to be documented also by Western blots to show changes of intensity of P-gp bands

3.       Several controversies in P-gp expression data and corresponding P-gp activities have to be clearly described and discussed  

4.       This research team published a paper: “Induction and activation of Pglycoprotein by dihydroxylated xanthones protect against the cytotoxicity of the Pglycoprotein substrate paraquat”. Here they were applied similar experimental design on study of oxygenated xanthones as potential P-gp activators.  It would be interesting to have information way authors aimed on study of effect these xantones in current paper. Comparison of effect of this two groups of xanthone derivate should be included to discussion. If exist any differences in silico evaluation of types of xanthones it should be clearly stated and explained.

Author Response

Thank you very much for the competent and helpful review of our manuscript entitled and please find in the attached file the answers to all the questions raised.

.

Reviewer 2 Report

Review of Manuscript ID: molecules-432332

Title: Newly synthetized oxygenated xanthones as potential P-glycoprotein activators – in vitro, ex vivo and in silico studies. Eva Martins, Vera Silva, Agostinho Lemos, Andreia Palmeira, Ploenthip Puthongking, Emília Sousa *, Carolina Rocha-Pereira, Carolina I. Ghanem, Helena Carmo, Fernando Remião *, Renata Silva *

The authors make a valid argument regarding the role P-glycoprotein plays in protecting cells from cytotoxic molecules. Specifically, they center their focus on PQ. Their logic follows that decreases in P-glycoprotein expression and activity would increase cytotoxicity, while increases in P-glycoprotein activity would decrease cytotoxicity. Since PQ is toxic and a P-gp substrate the premise is true. Given this, they evaluated a series (class) of molecules (oxygenated zanthones- OX-1 through OX-6) to determine their potential to induce P-glycoprotein activity and enhanced protection from PQ cytotoxicity. To accomplish this they performed the following experiments.

Characterized the OX1-6 molecular structure- Fig1

Evaluated the cytotoxicity of the potential inducers alone in Caco cells. (fig2)

Measured P-gp expression levels after 24 hrs exposure to OX1-6 (fig3).

Measured P-gp activity levels after short and longterm exposure to OX1-6 (fig4-5).

Compared cell death from increasing concentrations of PQ with OX1-6 (fig6) and evaluated the curves at EC 50.

To confirm that P-gp role in the protection they repeated the cell death curves with and without P-gp inhibition (Ela) Fig7.

Chose a candidate molecule OX-6 and tested Pgp-transport (inhibited and non-inhibited) in ex vivo in everted intestinal sacs (Fig8).

They transitioned their work to in silico and characterized the docking of the OXs in relationship to known inhibitor and inducers. (Fig 9-10).

In addition, they determined docking score and fit values of the OXs to P-gp induction and activation Pharmacophores (table 3-4).

The work is interesting, good and expansive, and the opinion of this reviewer is that it is worthy of publishing.  There are some concerns that should be addressed. There are a few sentences that will need editorial oversite and correction.  I will yield to the discretion of the editor to determine  mandatory rewvisions for publication.

Suggestions, questions, and concerns to be addressed.

Extend the OXs cytotoxicity at 50 uM out to 48 hours. Just to insure no toxicity occurs.

Does Inhibiting transcription and translation ablate the inductive effects of the OXs? This experiment would provide insight into mechanism in the early section of the work.

Western blots would serve well to complement Fig 3 to determine P-gp levels.

 Also, a KO P-gp (null) cell line as a true negative control should be used in figure 3. It would give the researchers a valid nonspecific background number to subtract from all the samples. This would increase the sensitivity of the assay and strengthen the data.

Do the authors know if the OX induction is reversible? How long does the induction last? This could be very important for clinical relevance going forward.

Does the induction by OXs occur in purified vesicles or purified membranes? This would further their findings that it is direct interaction (OX-1,2,5&6) and not a signaling event to an ancillary molecule involved in allosteric regulation and it would further confirm the in-silico work.

Lastly, it would be helpful if the authors could propose a mechanism beyond “direct interaction at a specific location” for how OX induces P-gp. Maybe a graphical depiction.

Author Response

Thank you very much for the competent and helpful review of our manuscript entitled and please find in the attached file the answers to all the questions raised.

Reviewer 3 Report

The authors assessed the P-gp modulatory effects of several oxygenated xanthones by using in vitro, ex vivo and in silico models. This research is well-designed and thoroughly undertaken. Authors have some interesting discovery and conclude that P-gp activity/expression could be modulated by several oxygenated xanthones. This manuscript is also generally well-written and suitable for publication in Molecules.

Minors:

Line 37-38, “an effect selectively blocked by zosuquidar, a P-gp inhibitor” may be changed into “which was selectively blocked by a model P-gp inhibitor zosuquidar”

Figure 4 and 5, authors need to clarify  1) if the oxygenated xanthones are auto-fluorescent and might interfere with the Rhodamine 123 signal; 2) the solvent used to dissolve the oxygenated xanthones and the final vehicle concentrations.

Table 1 and 2, authors should provide the 95% confidence of intervals for EC50

Author Response

(The authors gave the same response as above.)

Round 2

Reviewer 1 Report

I accept improvement made by authors in revised manuscript